# Antioxidant and Antiinflammatory Effects of *Epilobium parviflorum*, *Melilotus officinalis* and *Cardiospermum halicacabum* Plant Extracts in Macrophage and Microglial Cells

**DOI:** 10.3390/cells10102691

**Published:** 2021-10-08

**Authors:** Stefania Merighi, Alessia Travagli, Paola Tedeschi, Nicola Marchetti, Stefania Gessi

**Affiliations:** 1Department of Translational Medicine, University of Ferrara, 44121 Ferrara, Italy; mhs@unife.it (S.M.); alessia.travagli@edu.unife.it (A.T.); 2Department of Chemistry, Pharmaceutical and Agricultural Sciences, University of Ferrara, 44121 Ferrara, Italy; tdspla@unife.it (P.T.); mrcncl@unife.it (N.M.)

**Keywords:** plant extracts, human health, inflammation, oxidative damage, mechanism of action, A_2A_ adenosine receptor, phenols, flavonoids, tannins

## Abstract

Background: We investigated the phenolic content characterizing different plant extracts from *Epilobium parviflorum*, *Cardiospermum halicacabum*, and *Melilotus officinalis*, their antioxidant, antiinflammatory effects, and their mechanism of action. Methods: plant samples were macerated in 40% ethanol or hot/ cold glycerate and assessed for polyphenols content. The antioxidant activity was investigated by DPPH radical scavenging assay and H_2_DCFDA test in LPS-stimulated RAW264.7 macrophages and N9 microglial cells. MTS experiments and antiinflammatory properties verified cellular toxicity through NO assay. Interaction with A_2A_ adenosine receptors was evaluated through binding assays using [^3^H]ZM241385 radioligand. Results: Polyphenols were present in 40% ethanol plant extract, which at 0.1–10 µg/µL achieved good antioxidant effects, with a DPPH radical scavenging rate of about 90%. In LPS-stimulated cells, these plant extracts, at 1μg/μL, did not affect cell vitality, displayed significant inhibition of H_2_DCFDA and NO production, and inhibited ZM 241385 binding in CHO cells transfected with A_2A_ receptors. RAW 264.7 and N9 cells presented a density of them quantified in 60 ± 9 and 45 ± 5 fmol/mg of protein, respectively. Conclusion: *Epilobium parviflorum*, *Cardiospermum halicacabum,* and *Melilotus officinalis* extracts may be considered a source of agents for treating disorders related to oxidative stress and inflammation.

## 1. Introduction

Natural products, deriving from medicinal plants or herbs, are a source of numerous chemical compounds that regulate several biological functions and exert various beneficial effects on human health [1,2]. The World Health Organization has reported that 80% of the consumers worldwide are confident that plant medicines may be a valid option for their health and, indeed, thousands of plant species can offer strategies of intervention [3,4,5,6]. Indeed, a wide part of medical products commercially available has origin from plants [7]. However, even though there is a variety of plant resources in the natural world, only a limited percentage of them have been exploited and investigated from a scientific point of view. Their mechanism of action should be clarified to shed light on both beneficial and adverse effects. The therapeutic applications of natural products may cover a wide range of human diseases, where inflammation plays a crucial role, spanning from asthma, inflammatory arthropathies, diabetes, cancer, atherosclerosis, Parkinson’s, and Alzheimer’s diseases [8,9,10]. Inflammation is a complex response of the body’s immune system whose goal is to fight microbial infection and any tissue damage, thereby removing noxious stimuli and restoring normal cell physiology [9]. Important cells involved in the fight against inflammation are activated through exposure to bacterial lipopolysaccharides (LPS) [11] and modulate inflammatory mediators such as nitric oxide (NO) and free radicals [12]. A molecule regulating the inflammatory process is the endogenous and ubiquitous nucleoside, adenosine, exerting its functions by activating four G-protein-coupled receptors, named A_1_, A_2A_, A_2B_, and A_3_ adenosine receptors [13]. Among them, the A_2A_ subtype receives more attention as a therapeutic target to treat inflammation [14]. Specifically, in the brain, its activation is associated with proinflammatory and neurotoxic outcomes, whereas it is blocked with neuroprotection. The antagonism of the A_2A_ adenosine receptor is considered positive in different animal models of Huntington’s and Alzheimer’s disease, epilepsy, and excitotoxic conditions, including ischemia [15,16,17,18], and it is used as a treatment for Parkinson’s disease [19]. On the other hand, in peripheral blood cells, including neutrophils, lymphocytes, macrophages, its stimulation provides antiinflammatory effects protecting tissues from injury and stress [13,20,21]. The contradictory roles of A_2A_ in the brain vs. peripheral blood cells might be due to the stimulation of glutamate release with consequent cytotoxic effects at the neuronal level, in contrast to the increase of cAMP in blood cells playing a role in immunosuppression [13,14,17]. This study focused our attention on three plant extracts, including *Epilobium parviflorum*, *Melilotus officinalis*, and *Cardiospermum halicacabum*, whose beneficial properties are worldwide known for ages. *Epilobium parviflorum*, also known as a willow herb, is the most common species belonging to the Onagraceae and is exploited in the pharmaceutical, food, and cosmetic industries [22]. *Melilotus officinalis* is a plant of legume belonging to the family of Fabaceae, native in Eurasia, and is an annual herb. It is also known as yellow sweet clover. It was first published in the “European Pharmacopoeia” eighth edition, widely distributed worldwide [23]. *Cardiospermum halicacabum* is a herbal plant of the Sapindaceae family, native in India, Africa, and South America, used in Chinese medicine for a long time [24]. The properties of these plants, according to folk medicine, are various and include analgesic, antiinflammatory, antimicrobial, antioxidant, and antitumoral effects [24,25,26]. Despite the already existing information about the antiinflammatory properties of these traditional plants, pharmacological data supporting their therapeutic application alongside clinical research are required to evaluate their medical benefit. In fact, different studies focused their attention on analyzing and characterizing the active components of different extracts to discover new therapeutic molecules. However, there is still a lack of information about the molecular mechanism activated by the synergism of the whole extract. For these reasons, this study aimed to characterize, in two different models, including RAW 264.7 murine macrophages and N9 murine microglial cells, the antioxidant and antiinflammatory properties of the plant extracts prepared in different solvents, and to investigate, for the first time, the potential involvement of A_2A_ adenosine receptors in their mechanism of action.

## 2. Materials and Methods

### 2.1. Materials

Whatman GF/B glass fiber filters were from PerkinElmer (Milan, Italy). [^3^H]ZM 241385 was by Campro Scientific (Berlin, Germany). All other reagents were from Sigma Aldrich (Milan, Italy).

### 2.2. Plant Extracts

*Epilobium parviflorum*, *Melilotus officinalis,* and *Cardiospermum halicacabum* were kindly provided by Agripharma agricultural cooperative society (Padua, Italy). In detail, *Epilobium parviflorum* (Schreb.) (collected plant material from North Europe; voucher No.: BPLR070ATXA), *Melilotus officinalis, and Cardiospermum halicacabum* (cultivated plant material from Italy; voucher No.: L. MEL1809B and L. CARDI1806L, respectively) were studied. The dried aerial part of *Epilobium parviflorum*, aerial flower part of *Melilotus officinalis,* and flowering tops of *Cardiospermum halicacabum* contain the plants’ main active constituents from literature data [27,28,29], were obtained through low-temperature drying. Then, they were shredded and then macerated in 40% *v*/*v* ethanol or hot or cold glycerate with euxil 9010, for 21 days, at room temperature, in dark conditions. A ratio of 1:10 and 1:3 (g over solvent volume, mL) was used for 40% *v*/*v* ethanol and hot/cold glycerate extracts, respectively. Then, the thick mass of 40% *v*/*v* ethanol extracts was filtered several times through tangential flow microfiltration with a ceramic filter, having a porosity of 0.2 µm diameter. At the same time, hot or cold glycerate extracts through a paper filter with porosity of 8–20 µm diameter. Finally, the obtained liquid part, about 90%, was bottled at cold temperatures.

### 2.3. Total Phenolic Content

Total phenolic content was determined using the classic Folin Ciocalteu colorimetric method described in Reference [30], partially modified. Then, 500 μL of Folin–Ciocalteu reagent were added to 25 μL of extract. The mixture was allowed to stand for 5 min, and then 2 mL of a 10% aqueous Na_2_CO_3_ solution was added. The final volume was adjusted to 10 mL. Samples were allowed to stand for 90 min at room temperature before measurement at 700 nm vs. the reagent blank, using a Beckman DU730 UV-vis spectrophotometer. The amount of total phenolics is expressed as gallic acid equivalents (µg gallic acid/µL of plant extracts) through the calibration curve. The calibration curve range was 0.5–10 ppm.

### 2.4. Flavonoid Content

Total flavonoid content was determined using a colorimetric method. Where 150 μL of 5% NaNO_2_ solution was added to 25 μL of plant extract and allowed to stand for 5 min, and then 300 μL of 10% AlCl_3_ solution and 1 mL of NaOH 1M were added. The final volume was adjusted to 5 mL, and the absorption was measured at 510 nm vs. the reagent blank, using a Beckman DU730 UV-vis spectrophotometer. The amount of flavonoids is expressed as (+)-catechin equivalents (µg (+)-catechin/μL of plant extracts). The calibration curve range was 1–10 ppm.

### 2.5. Total Condensed Tannins

The determination of total condensed tannins was obtained using the colorimetric method described in [31], partially modified. 3 mL of vanillin (4% in MeOH, *w*/*v*) and 1.50 mL of HCl were added to 25 μL of plant extracts. The final volume was then adjusted to 5 mL with methanol, and the absorption was measured at 500 nm vs. the reagent blank. The amount of total condensed tannins was expressed as (+)-catechin equivalents (µg (+)-catechin/*µ*L of plant extracts) through the calibration curve of (+)-catechin. The calibration curve considered was between 0.5–10 ppm.

### 2.6. Cell Cultures

RAW 264.7 macrophage murine cells (BS TCL 177, IZSLER Biobank, Brescia, Italy) were cultured in Dulbecco’s Modified Eagle’s Medium (DMEM)–high glucose, supplemented with 10% of heat-inactivated fetal bovine serum (FBS) and 1% of a penicillin (100 U/mL) and streptomycin (100 μg/mL) solution. N9 murine microglial cells were grown in Iscove’s Modified Dulbecco’s Medium (IMDM) with 5% heat-inactivated Australian FBS, 1% penicillin, and streptomycin, kindly provided by Prof. Ricciardi-Castagnoli. CHO cells (American Tissue Culture Collection, ATCC, Manassas, VA, USA) transfected with human A_2A_ adenosine receptor (hA_2A_CHO) [32] were maintained in DMEM with nutrient mixture F12 without nucleosides, summed with 10% fetal calf serum, penicillin (100 U/mL), streptomycin (100 mg/mL), l-glutamine (2 mM), and Geneticin (G418, 0.2 mg/mL). Cells were kept in a humidified environment with 5% CO_2_ and 37 °C of temperature and were diluted three times a week to maintain the optimal confluence (80%).

### 2.7. Cellular Treatments

RAW 264.7 and N9 cell lines were stimulated with 1 µg/mL of lipopolysaccharide (LPS) (from Escherichia coli, serotype 055:B5, soluble in cell culture medium) for 24 h to trigger the proinflammatory response. Other treatments consisted of different concentrations (2.5 µg/µL, 1 µg/µL, and 0.1 µg/µL) of the plant extracts, added 30 min before LPS. Before every experiment, the cell medium was changed with serum-free medium.

### 2.8. DPPH Test

The antioxidant capacity of different concentrations of 40% ethanol, hot and cold glycerate plant extracts was tested with a 2,2-diphenyl-1-picrylhydrazyl (DPPH) assay. In detail, each tested extract and the ascorbic acid were added, in duplicate, in a black 96 well-plate containing 0.1 mM DPPH or methanol for the blank. The 96 well-plate was mixed for 30 min in an orbital shaker in the dark at room temperature. Then, the absorbance was measured with the Ensight multimodal plate reader (Perkin Elmer, Milan, Italy) at 517 nm. The antioxidant ability was calculated as a percentage of inhibition vs. control obtained in the absence of extract, while ascorbic acid (50 µM) was used as a positive control. The IC_50_ values were calculated as the concentration of sample required to scavenge 50% of DPPH free radicals.

### 2.9. MTS Assay

The MTS assay was performed to determine cells vitality according to the manufacturer’s protocol from the CellTiter 96 AQueous One Solution cell proliferation assay (Promega, Milan, Italy). Cells were plated in 96-multiwell plates (30,000 cells/well), allowed to attach overnight, then 100 μL of complete medium was added to each well in the absence and the presence of 40% ethanol plant extracts for 24 h. At the end of the incubation period, MTS solution was added to each well. The optical density of each well was read on a spectrophotometer at 570 nm.

### 2.10. H_2_DCFDA Assay

The antioxidant potential of 40% ethanol plant extracts was tested in RAW 264.7 macrophage and N9 microglial cells by the 2′,7′-dichlorofluorescein diacetate (H_2_DCFDA) assay. In detail, 30,000 cells were seeded in a black 96 well plate and incubated overnight. Subsequently, treatments were performed in a serum-free medium. After 24 h, the supernatant of each well was removed, and 100 µL of 10 µM H_2_DCFDA solution was added. The plate was then incubated in the dark at 37 °C. After 1 h, three PBS washes were performed, and then 100 µL of PBS was added to each well. The fluorescence was read with the Ensight multimodal plate reader at an excitation of 485 nm and an emission of 538 nm (Perkin Elmer, Milan, Italy).

### 2.11. Nitric Oxide Assay

The antiinflammatory potential of 40% ethanol plant extracts was tested in RAW 264.7 and N9 microglial cells with the Nitrate/Nitrite Colorimetric Assay Kit purchased by Vinci Biochem (Florence, Italy). In detail, 150,000 cells were seeded in a 24 wells plate and incubated for 24 h; 80 μL of the supernatants of each well were transferred to a 96 well plate with 10 μL of the nitrate reductase and 10 μL of its cofactor. After 2 h of incubation, the two Griess reagents were added, converting the total nitrite to a purple azoic compound. The absorbance measurement was performed with the Ensight multimodal plate reader (Perkin Elmer, Milan, Italy) set at 550 nm. The standard curve was performed with nitrate, allowing the determination of the nitrate + nitrite concentration, which is proportional to the red absorbance.

### 2.12. Membrane Preparation

After medium removal and a wash step with PBS, hA_2A_CHO, RAW 264.7 macrophage, and N9 microglial cells were harvested in a cold hypotonic buffer. The solution was homogenized with a Polytron and centrifuged at 18,000 rpm for 30 min at 4 °C. Cells were resuspended in A_2A_ buffer (50 mM Tris-HCl buffer, pH 7.4, containing 1 mM EDTA and 10 mM MgCl_2_) with 3 U.I./mL adenosine deaminase (ADA), incubated at 37 °C for 30 min to remove endogenous adenosine. Protein concentration was measured by the method of Biorad using bovine serum albumin as a standard.

### 2.13. Radioligand Binding Experiments

[^3^H]ZM 241385 (specific activity 20 Ci/mmol), a potent and selective A_2A_ receptor ligand, was used in both saturation and competition binding experiments [32]. In detail, different concentrations of plant extracts in 40% ethanol were incubated, in duplicate, in glass tubes containing membranes from hA_2A_CHO, A_2A_ buffer, and 1 nM [^3^H]ZM 241385. To determine non-specific binding, 1 µM of ZM 241385 was added. For saturation experiments increasing concentrations of [^3^H]ZM 241385 (0.1–10 nM) were incubated with membranes from RAW 264.7 and N9 cells. After 1 h at 4 °C, bound and free radioactivity were separated through a filtration method with Brandel Whatman using GF/B glass fiber filters (Brandel Instrument, MD, USA). The radioactivity was quantified by a Tri-Carb Packard 2500 TR scintillation counter (Perkin-Elmer Life and Analytical Sciences, Boston, MA, USA).

### 2.14. Statistical Analysis

The values in the figures are expressed as mean ± standard error (SEM) of three independent experiments. When required, data sets were examined by one-way analysis of variance (ANOVA) and Dunnett’s test as post analysis. A *p*-value less than 0.05 was considered statistically significant.

## 3. Results

### 3.1. Phenols Contents of Plant Extracts

Quantitative phenolics data, expressed as µg/µL of plant extracts, are shown in Table 1.

A noteworthy difference in total phenolic content between the three different plants was observed, with the *Epilobium parviflorum* sample being the richest one. Among the three types of extraction, the highest phenolics content was revealed only for the 40% ethanol *Cardiospermum halicacabum* plant extracts. Among the different extractions, the flavonoid content in ethanol extract was similar to the glycerate ones for *Epilobium parviflorum* and *Melilotus officinalis*. At the same time, condensed tannins were present in lower concentrations in 40% ethanol plant extracts.

### 3.2. Antioxidant Properties of Plant Extracts

Nine plant extracts were investigated for their antioxidant properties by the DPPH assay. In detail, different concentrations of *Epilobium parviflorum*, *Melilotus officinalis,* and *Cardiospermum halicacabum* extracts, in their own solvent, hot glycerate, cold glycerate, and 40% ethanol, respectively, were investigated. The results, expressed as % of inhibition of DPPH activity and the respective IC_50_ values, are presented in Table 2.

The antioxidant ascorbic acid 50 µM was used as an internal positive control in each experiment and was always able to reduce DPPH absorbance by 85 ± 7%. All the plant extracts showed an inverse proportionality between their concentration and the percentage of inhibition of the radical DPPH, with no antioxidant activity when diluted at 0.04 µg/µL and 0.01 µg/µL, for glycerate and 40% ethanol extracts, respectively, except for *Epilobium parviflorum* 40% ethanol which showed a 40 ± 5% of DPPH inhibition. Among the three types of extraction, the highest DPPH radical scavenging activity was generally revealed for the 40% ethanol plant extracts, as revealed by IC_50_ values. Specifically, *Epilobium parviflorum*, the most potent natural extract, showed its significant antioxidant properties when diluted to 10 µg/µL, 1 µg/µL and 0.1 µg/µL, as revealed by the inhibition of DPPH absorbance at 517 nm, of 92 ± 6, 90 ± 5 and 81 ± 6%, respectively. *Melilotus officinalis* inhibited DPPH of 90 ± 1 and 86 ± 2% at 10 µg/µL and 1 µg/µL concentration, respectively, while the effect was reduced to 30 ± 3% with 0.1 µg/µL concentration. *Cardiospermum halicacabum* reduced DPPH absorbance of 89 ± 4% and 82 ± 3% at 10 µg/µL and 1 µg/µL concentration, respectively, and showed a minimum effect of 26 ± 2% inhibition at 0.1 µg/µL concentration.

### 3.3. Cells Viability Following Treatment with Epilobium parviflorum, Melilotus officinalis and Cardiospermum halicacabum on RAW 264.7 Macrophage and N9 Microglial Cells

The effects of plant extracts on cell viability were investigated in RAW 264.7 macrophages and N9 microglial cells, chosen as models of cells involved in peripheral and central inflammation, respectively. In particular, as the better antioxidant effects on DPPH reduction were observed with extracts prepared in 40% ethanol, we evaluated their potential toxicity using MTS assay. In order to start with a nontoxic concentration of ethanol extract, we treated cells with the following plant extracts concentration 2.5 µg/µL, 1 µg/µL, 0.1 µg/µL. Our results showed that *Cardiospermum halicacabum* 2.5 µg/µL reduced cell viability of both N9 and RAW cells, while *Epilobium parviflorum* 2.5 µg/µL was toxic in RAW cells, no toxicity was observed for all the other samples (Table 3). Therefore, the concentrations 1 µg/µL and 0.1 µg/µL were used in the next experiments for all the plant extracts investigated.

### 3.4. Antioxidant Properties of Epilobium parviflorum, Melilotus officinalis, and Cardiospermum halicacabum on RAW 264.7 Macrophage Cells

Ethanol plant extracts of *Epilobium parviflorum*, *Melilotus officinalis,* and *Cardiospermum halicacabum*, with a powerful antioxidant potential but without toxic effect, were chosen to be tested in a cellular model of peripheral and central inflammation, represented by macrophage RAW 264.7 and N9 microglial cells stimulated with LPS 1 µg/mL, a known proinflammatory mediator. In detail, the ability of herbal extracts to prevent oxidative damage was verified by H_2_DCFDA assay. As shown in Figure 1A,B, none of them, when used alone at 1 µg/µL concentration, significantly modified the H_2_DCFDA oxidation of control cells in RAW 264.7 and N9, respectively. Then, the antioxidant potential of the three plant extracts was investigated in the presence of LPS in RAW and N9 cells. *Epilobium parviflorum*, *Melilotus officinalis,* and *Cardiospermum halicacabum* at 1 µg/µL and 0.1 µg/µL concentrations were able to significantly decrease the H_2_DCFDA absorbance increased by LPS in macrophage. In contrast, in N9 cells, only 1 µg/µL plant extracts concentrations showed a significant effect (Figure 1C,D, respectively). These results indicate that plant extracts investigated tend to be more potent in macrophages than in microglial cells.

### 3.5. Antiinflammatory Properties of Epilobium parviflorum, Melilotus officinalis and Cardiospermum halicacabum on RAW 264.7 and N9 Cells

The antiinflammatory properties of *Epilobium parviflorum*, *Melilotus officinalis,* and *Cardiospermum halicacabum* ethanol plant extracts were tested on RAW 264.7 macrophage and N9 microglial cells by NO assay. Firstly, to investigate the effect of herbal extracts on basal NO production, cells were treated with 40% ethanol *Epilobium parviflorum*, *Melilotus officinalis,* and *Cardiospermum halicacabum* diluted 1 µg/µL. As shown in Figure 2A,B, none of them alone significantly modified the NO produced by RAW 264.7 and N9 cells, respectively. Then, the antiinflammatory activity of these plant extracts was investigated treating the cells with different concentrations of *Epilobium parviflorum*, *Melilotus officinalis,* and *Cardiospermum halicacabum* (1 µg/µL and 0.1 µg/µL) in combination with 1 μg/mL LPS. As expected, LPS treatment of the cells for 24 h increased NO secretion in RAW 264.7 and N9 cells, reaching a concentration of 31 ± 7 and 65 ± 9 µM, respectively. *Epilobium parviflorum* and *Cardiospermum halicacabum* 1 µg/µL were able to significantly decrease LPS-stimulated NO production, suggesting a strong anti-inflammatory potential of these plant extracts in both cell lines. As for 0.1 µg/µL concentration of both, a different behavior was observed in RAW 264.7 cells where the effect was still present (45 ± 5% and 32 ± 4% of inhibition, respectively) in contrast to N9 cells where no reduction was detected. *Melilotus officinalis* significantly reduced NO secretion when diluted 1 µg/µL; however, its antiinflammatory potential was lost when diluted 0.1 µg/µL in both cell lines (Figure 2C,D).

### 3.6. Affinity of Epilobium parviflorum, Melilotus officinalis, and Cardiospermum halicacabum for A_2A_ Adenosine Receptors

Finally, to evaluate whether the antioxidant and antiinflammatory action of the plant compounds was due to the A_2A_ Adenosine Receptors, known for their role in the antiinflammatory process, competition binding experiments using the selective and high-affinity radioligand antagonist [^3^H]ZM 241385 were performed in hA_2A_CHO. In detail, different concentrations of *Epilobium parviflorum*, *Melilotus officinalis,* and *Cardiospermum halicacabum* 40% ethanol extracts (10 µg/µL and 1 µg/µL) were compared to unlabelled ZM 241385 1 µM. As shown in Figure 3, *Epilobium parviflorum* 10 µg/µL and 1 µg/µL significantly reduced the [^3^H]ZM 241385 binding to A_2A_ adenosine receptors of 40 ± 5% and 19 ± 3%, respectively. The 10 µg/µL *Melilotus officinalis* displaced 86 ± 9% of the [^3^H]ZM 241385 binding, suggesting a very high affinity of this compound for A_2A_ adenosine receptors. The effect was still significant with the 1 µg/µL concentration (49 ± 6%). Finally, the binding inhibition of *Cardiospermum halicacabum* was higher when diluted 1 µg/µL (47 ± 5%) in comparison to when diluted 10 µg/µL concentration (20 ± 4%) (Figure 3).

### 3.7. Expression of A_2A_ Adenosine Receptors in RAW 264.7 and N9 Cells

To evaluate the expression of A_2A_ adenosine receptors in RAW 264.7 and N9 cells, binding experiments with [^3^H]ZM 241385 were carried out. In particular, cellular membranes obtained from RAW 264.7 and N9 cells were incubated with increasing concentrations of the selective and high-affinity radioligand antagonist in the range 0.1–10 nM for 1 h. Following the filtration procedure to separate bound and free radioactivity, the results revealed that these cells presented A_2A_ adenosine receptors with a density of 60 ± 9, 45 ± 5 fmol/mg of proteins and [^3^H]ZM 241385 had an affinity value of 0.9 ± 0.2 and 1.2 ± 0.2 nM to them, respectively.

## 4. Discussion

Natural products exert a relevant role in human health as well as in drug discovery [33]. Extracts of three commercially available medicinal plants *Epilobium parviflorum*, *Melilotus officinalis,* and *Cardiospermum halicacabum* are used as herbal medicines in several countries for their antioxidant and antiinflammatory effects. Still, their mechanism of action is not fully understood. For the present research, specific parts of them were selected. In particular, the dried aerial part of *Epilobium parviflorum*, aerial flower part of *Melilotus officinalis,* and flowering tops of *Cardiospermum halicacabum* were chosen based on the richest parts containing bioactive substances [27,28,29]. First of all, we investigated the antioxidant properties of *Epilobium parviflorum*, *Melilotus officinalis*, and *Cardiospermum halicacabum* plants through the DPPH assay. The solvent significantly affects the phytochemical pattern of a given herbal extract. Because the chemistry modulates the biological activity, different solvent systems were used, including hot and cold glycerate and 40% ethanol, to prepare plant extracts. By comparing the different types of extraction, all the plant extracts demonstrated a good radical scavenging activity. Still, the most significant effect was observed with 40% ethanol plant extracts, and the percentage of inhibition correlated with extracts concentration. It is well known that ethanol is a polar solvent able to extract a significant amount of polyphenolic compounds like the flavonoids and tannins responsible for the observed antiradical activity of plant extracts used in this study [34,35]. Indeed, as for *Cardiospermum halicacabum*, the 40% ethanol extracts contained higher amounts of both polyphenols and flavonoids in comparison to hot and cold glycerate fractions. As for *Epilobium parviflorum* and *Melilotus officinalis,* these chemical classes of compounds were not significantly more concentrated in 40% ethanol extracts, suggesting that other non-flavonoid components may be responsible for the higher antioxidant activity [25]. Therefore, we evaluated their antioxidant and antiinflammatory properties in RAW 264.7 macrophages and N9 microglial cells, chosen as in vitro cellular models of peripheral and neuroinflammation, respectively. Importantly, when cell vitality was evaluated, *Epilobium parviflorum* and *Cardiospermum halicacabum* 40% ethanol plant extracts showed toxic effects in RAW 264.7 and N9 cell lines, respectively, when tested at 2.5 µg/µL, but were safe at 1 µg/µL and 0.1 µg/µL concentrations. On this basis, 1 µg/µL and 0.1 µg/µL concentrations were chosen for the next experiments to evaluate their ability to reduce radical oxygen and NO production in in vitro cellular models. The activation of macrophages and microglia by the bacterial surface molecule LPS leads to the production of free oxygen and NO radicals, which exert important roles in inflammation, affecting many age-related diseases such as Alzheimer’s pathology [36]. As the antioxidant effects of natural extracts play an important role in reducing inflammation, we showed that all 40% ethanol plant extracts did not affect free radical production when tested alone. Still, they were able to potently counteract LPS-induced oxidative stress at both 1 µg/µL and 0.1 µg/µL concentrations. In addition, they were investigated for their ability to contrast inflammation, by evaluating their effect on NO production. Indeed, NO is a crucial signaling molecule playing a role in different biological activities, including immune and vascular function. Specifically, it activates immune cells and, in particular, macrophages to induce a protective response. Still, its excessive secretion is responsible for brain damage in neurodegenerative diseases and ischemia, suggesting that its modulation is necessary to preserve human health [37,38]. Therefore, it is important to find new modulators of NO production, and natural products may be potential leaders as antiinflammatory mediators [38]. Our results show that all 40% ethanol plant extracts could reduce NO production in both cell lines investigated. In particular, we observed that *Epilobium parviflorum* and *Cardiospermum halicacabum*, at 0.1 µg/µL concentration, reduced free radical and NO production only in RAW 264.7 cells, confirming their ability to cope up with oxidative stress, according to literature data [23,39,40]. In order to elucidate the mechanism of action of these 40% ethanol plant extracts, we evaluated their interaction with the A_2A_ adenosine receptor subtype, having a crucial role in reducing inflammation [41,42]. Indeed, it was demonstrated that A_2A_ receptor-deficient mice presented increased inflammation and tissue damage in models of acute liver injury, endotoxin-associated sepsis, and infected wound model, suggesting a non-redundant role in the attenuation of inflammation and tissue damage in vivo [43,44]. Targeting these receptors by selective agonists or natural products may lead to better protocols of antiinflammatory treatments [45]. As an example of compounds interacting with A_2A_ adenosine receptors to produce beneficial effects, caffeine and resveratrol have been described [46,47]. Interestingly, we found that all three 40% ethanol plant extracts were able to compete with the selective A_2A_ antagonist radioligand ZM 241385, in CHO cells transfected with A_2A_ adenosine receptors, being *Melilotus officinalis* the most potent extract, suggesting their interaction with this membrane receptor subtype. Therefore, radioligand binding experiments demonstrated the expression of A_2A_ adenosine receptors in both RAW 264.7 and N9 cells, with a density of 60 ± 9 and 45 ± 5 fmol/ mg of protein, as potential targets of *Epilobium parviflorum*, *Melilotus officinalis*, and *Cardiospermum halicacabum* to counteract inflammation. In conclusion, the results of this study show that the ethanolic extracts from the dried aerial part of *Epilobium parviflorum*, aerial flower part of *Melilotus officinalis*, and flowering tops of *Cardiospermum halicacabum* are characterized by the presence of several polyphenols, in particular flavonoids and condensed tannins, and may be considered as a potential source of agents for the treatment of disorders related to oxidative stress and inflammation.

## Figures and Tables

**Figure 1 cells-10-02691-f001:**
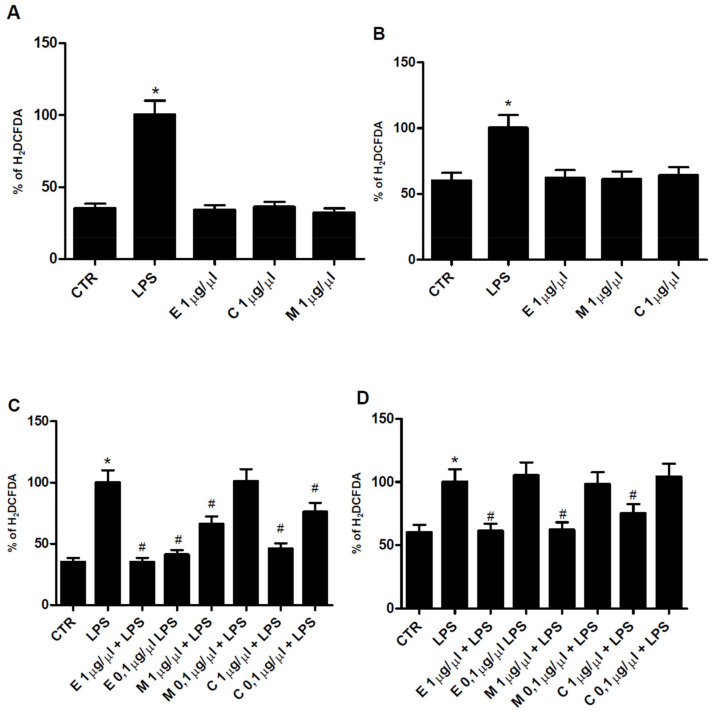
ROS inhibition by 40% ethanol plant extracts. Effect of *Epilobium parviflorum*, *Melilotus officinalis,* and *Cardiospermum halicacabum* 1 µg/µL on ROS (H_2_DCFDA) production in RAW 264.7 macrophage (**A**) and microglial N9 (**B**) cells. Effect of 1 µg/µL and 0.1 µg/µL *Epilobium parviflorum*, *Melilotus officinalis* and *Cardiospermum halicacabum* on ROS (H_2_DCFDA) production in LPS(1 µg/mL)-treated RAW 264.7 macrophage (**C**) and microglial N9 cells (**D**). Bars represent % mean ± SEM. * *p* < 0.05 vs. control; # *p* < 0.05 vs. LPS.

**Figure 2 cells-10-02691-f002:**
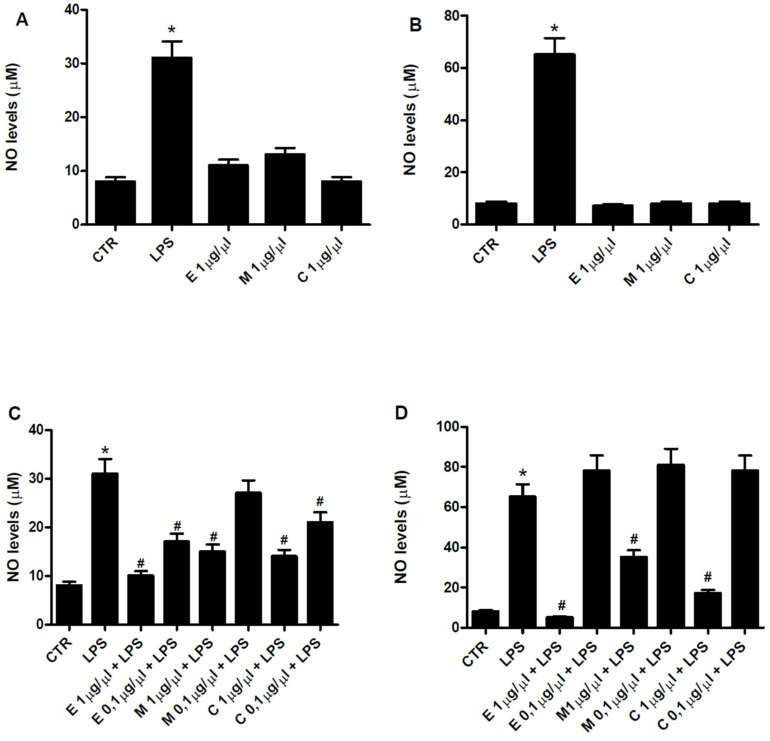
NO inhibition by 40% ethanol plant extracts. Effect of *Epilobium parviflorum*, *Melilotus officinalis,* and *Cardiospermum halicacabum* 1 µg/µL on NO levels in RAW 264.7 macrophage (**A**) and N9 microglial cells (**B**). Effect of *Epilobium parviflorum*, *Melilotus officinalis* and *Cardiospermum halicacabum* 1 µg/µL and 0.1 µg/µL on NO production in LPS(1 µg/mL)-treated RAW 264.7 macrophage (**C**) and N9 microglial cells (**D**). Bars represent mean ± SEM. * *p* < 0.05 vs. control; # *p* < 0.05 vs. LPS.

**Figure 3 cells-10-02691-f003:**
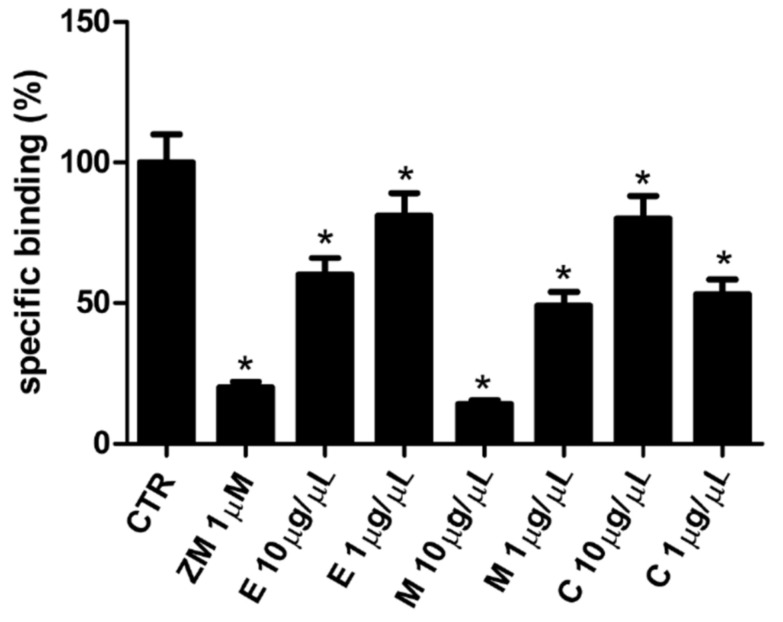
A_2A_ receptor binding by 40% ethanol plant extracts. Radioligand competition binding assay to human A_2A_ adenosine receptor transfected in CHO cells. Effect of 1 µM A_2A_ receptor antagonist ZM 241385, 1 µg/µL and 0.1 µg/µL *Epilobium parviflorum*, *Melilotus officinalis*, *Cardiospermum halicacabum* ethanol plant extracts on [^3^H]ZM 241385 specific binding. Bars represent % mean ± SEM. * *p* < 0.05 vs. control (CTR).

**Table 1 cells-10-02691-t001:** Polyphenols, flavonoids, and tannins content of the plant extract *Epilobium parviflorum*, *Melilotus officinalis,* and *Cardiospermum halicacabum* were prepared in hot and cold glycerate and 40% ethanol.

Plant Extracts	Solvent	Polyphenols (µg/µL Gallic Acid Eq.)	Flavonoids (µg/µL +(-) Catechin Eq.)	Tannins (µg/µL +(-) Catechin Eq.)
*Epilobium parviflorum*	hot glycerate	14.16 ± 0.04	4.78 ± 0.11	1.04 ± 0.01
	cold glycerate	14.85 ± 0.14	3.71 ± 0.01	1.43 ± 0.01
*Melilotus officinalis*	hot glycerate	4.12 ± 0.06	1.57 ± 0.01	0.76 ± 0.07
	cold glycerate	5.53 ± 0.07	2.66 ± 0.01	0.88 ± 0.09
*Cardiospermum halicacabum*	hot glycerate	2.82 ± 0.02	2.76 ± 0.06	1.48 ± 0.02
	cold glycerate	2.78 ± 0.03	2.08 ± 0.02	1.12 ± 0.01
*Epilobium parviflorum*	40% ethanol	16.79 ± 0.16	4.45 ± 0.04	0.56 ± 0.06
*Melilotus officinalis*	40% ethanol	3.18 ± 0.03	2.74 ± 0.02	0.16 ± 0.02
*Cardiospermum halicacabum*	40% ethanol	7.82 ± 0.07	5.28 ± 0.05	1.02 ± 0.01

Results are expressed as µg gallic acid equivalents/µL of plant extracts for polyphenols quantifications, µg (+)- catechin equivalents/µL of plant extracts for flavonoids and condensed tannins ± SEM.

**Table 2 cells-10-02691-t002:** Antioxidant effect of different dilutions of the plant extracts *Epilobium parviflorum*, *Melilotus officinalis*, and *Cardiospermum halicacabum* prepared in hot and cold glycerate and 40% ethanol.

Plant Extracts	Solvent	40 µg/µL	4 µg/µL	0.4 µg/µL	0.04 µg/µL	IC_50_ (µg/µL)
*Epilobium parviflorum*	hot glycerate	69 ± 7 *	63 ± 4 *	51 ± 3 *	13 ± 1	0.195 ± 0.022
	cold glycerate	72 ± 2 *	71 ± 1 *	61 ± 3 *	21 ± 3	0.117 ± 0.021
*Melilotus officinalis*	hot glycerate	70 ± 5 *	67 ± 6 *	24 ± 4 *	5 ± 1	0.141 ± 0.013
	cold glycerate	89 ± 4 *	74 ± 7 *	41 ± 5 *	7 ± 3	0.510 ± 0.053
*Cardiospermum halicacabum*	hot glycerate	65 ± 5 *	60 ± 3 *	24 ± 7 *	15 ± 2	0.892 ± 0.080
	cold glycerate	84 ± 9 *	61 ± 4 *	39 ± 3 *	8 ± 1	0.587 ± 0.075
		**10 µg/µL**	**1 µg/µL**	**0.1 µg/µL**	**0.01 µg/µL**	
*Epilobium parviflorum*	40% ethanol	92 ± 6 *	90 ± 5 *	81 ± 6 *	40 ± 5 *	0.014 ± 0.013
*Melilotus officinalis*	40% ethanol	90 ± 1 *	86 ± 2 *	30 ± 3 *	9 ± 1	0.227 ± 0.025
*Cardiospermum halicacabum*	40% ethanol	89 ± 4 *	82 ± 3 *	26 ± 2 *	16 ± 1	0.290 ± 0.027

Results are expressed as % ± SEM of inhibition of the 0.1 mM oxidant radical DPPH. * *p* < 0.05 vs. control.

**Table 3 cells-10-02691-t003:** Effect on cell viability of different dilutions of the plant extracts *Epilobium parviflorum*, *Melilotus officinalis*, and *Cardiospermum halicacabum* prepared in 40% ethanol on N9 microglial and RAW cells.

Plant Extracts	N9			RAW 264.7		
	2.5 µg/µL	1 µg/µL	0.1 µg/µL	2.5 µg/µL	1 µg/µL	0.1 µg/µL
*Epilobium parviflorum*	95 ± 6	98 ± 7	102 ± 8	33 ± 4 *	80 ± 9	99 ± 8
*Melilotus officinalis*	97 ± 9	98 ± 8	101 ± 11	96 ± 8	98 ± 9	105 ± 9
*Cardiospermum halicacabum*	69 ± 9 *	95 ± 3	97 ± 8	31 ± 4 *	98 ± 8	101 ± 9

Results are expressed as % ± SEM of control. * *p* < 0.05 vs. control.

## Data Availability

Not applicable.

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
