# Peer review of "Antioxidant and Antiinflammatory Effects of Epilobium parviflorum, Melilotus officinalis and Cardiospermum halicacabum Plant Extracts in Macrophage and Microglial Cells"

_cells, 2021, doi:10.3390/cells10102691_

Round 1
Reviewer 1 Report
In this study, the phenolic contents of plant extracts of Epilobium, Cardiospermum and Melilotus were assessed, and the antioxidant and anti-inflammatory properties of the extracts were evaluated in vitro. The extracts had potent antioxidant actions and suppressed inflammation in LPS-stimulated RAW264.7 macrophages and N9 microglial cells, at least in part through antagonism of A2A receptor. The authors conclude that Cardiospermum and Melilotus extracts possess agents with the potential to treat oxidative stress and inflammation linked disorders.
While the data presented is good and compelling, the extracts used are ill-defined. They do contain polyphenols, flavonoids, and tannins but these appear to account for only a small proportion of the extracted material. What is the rest? Does it contribute to bioactivity? The description of the procedures used in extract preparation is very unclear. It would be very difficult for others to reproduce these studies. This means the study is done with extracts prepared in an ill-defined way and having protective effects possibly due to unknown substances. The extraction procedures and extract compositions need to be greatly clarified. Were extractions based on results of prior work? If so, this should be cited.
Ln 43 ‘replacement’ Should this be ‘restoration’?
Ln 44 Something is missing. ‘against inflammation are ---- which can be activated through exposure to bacteria’
Ln 52-57 Are the contradictory roles of A2A in brain disease and anti-inflammatory blood cell activity agonist/antagonist- or dose-dependent? A little discussion on this would better put the significance of findings with the extracts in context.
Ln 58 Why did you select Epilobium, Melilotus and Cardiospermum?
Ln 85-92 See general comments. Much more detail of procedures used is needed or if based on literature this must be cited. Were extractions really done for one month? Under what conditions?
Ln 88-89 Why were the aerial part of Epilobium, aerial flower part of Melilotus and flowering tops of Cardiospermum used for extraction? Are these the plant parts used in folk medicine?
Ln 94 ‘27Singleton and Rossi (1965)’. The number has carried over with reference.
Ln 110 ‘28Broadhurst and Jones (1978)’. As above.
Ln 121 1% antibiotics as in ln 119?
Ln 129-133 Were extracts added along with LPS or after LPS activation?
Ln 133 How long before the study was media changed?
Ln 241 ‘ethanol, we’. Substitute ‘ethanol extract, we’.
Ln 244 ‘RAW cells, any toxicity’. Substitute ‘RAW cells, no toxicity’
Ln 260-262 Where is data for extracts alone. Not in Figures 1A and B, which is for LPS treated. Move ref to Figure 1A and B further down text and leave text on extracts alone as a statement (data not shown).
Ln 268 ‘investigated are more potent’. investigated tend to be more potent.
Ln 283-284 Where is data for extracts alone. Not in figure 2A and 2B, which is for LPS treated. Move ref to Figure 2A and 2B further down text and leave text on extracts alone as a statement (data not shown).
Ln 341-344 References for this observation?
Ln 402 ‘to contrast inflammation’ ‘to counteract inflammation’?
Author Response
Comments and Suggestions for Authors
In this study, the phenolic contents of plant extracts of Epilobium, Cardiospermum and Melilotus were assessed, and the antioxidant and anti-inflammatory properties of the extracts were evaluated in vitro. The extracts had potent antioxidant actions and suppressed inflammation in LPS-stimulated RAW264.7 macrophages and N9 microglial cells, at least in part through antagonism of A2A receptor. The authors conclude that Cardiospermum and Melilotus extracts possess agents with the potential to treat oxidative stress and inflammation linked disorders.
While the data presented is good and compelling, the extracts used are ill-defined. They do contain polyphenols, flavonoids, and tannins but these appear to account for only a small proportion of the extracted material. What is the rest? Does it contribute to bioactivity? The description of the procedures used in extract preparation is very unclear. It would be very difficult for others to reproduce these studies. This means the study is done with extracts prepared in an ill-defined way and having protective effects possibly due to unknown substances. The extraction procedures and extract compositions need to be greatly clarified. Were extractions based on results of prior work? If so, this should be cited.
We thank the reviewer for his/her kind comments and suggestions on our work. The extractions procedures were not based on the results of prior work so we have improved their description, as reported below.
Ln 43 ‘replacement’ Should this be ‘restoration’?
Done.
Ln 44 Something is missing. ‘against inflammation are ---- which can be activated through exposure to bacteria’
Done
Ln 52-57 Are the contradictory roles of A2A in brain disease and anti-inflammatory blood cell activity agonist/antagonist- or dose-dependent? A little discussion on this would better put the significance of findings with the extracts in context.
Ln 57 We have added the following comment according to the referee’s suggestion: “The contradictory roles of A2A in brain versus peripheral blood cells, may be due to the stimulation of glutamate release with consequent cytotoxic effects at neuronal level, in contrast to the increase of cAMP in blood cells playing a role in immunosuppression [13, 14, 17].
Ln 58 Why did you select Epilobium, Melilotus and Cardiospermum?
We selected those species because they are widely used in folk medicine due to their broad range of biological activities. Thus, despite the already existing information about the anti-inflammatory properties of the species, pharmacological data supporting their therapeutic application alongside clinical research are required to evaluate their medical benefit. We expressed this concept at line 70:
“The properties of these plants, according to folk medicine, are various and include analgesic, antiinflammatory, antimicrobial, antioxidant and antitumoral effects [24-26]. Despite the already existing information about the anti-inflammatory properties of these traditional plants, pharmacological data supporting their therapeutic application alongside clinical research are required to evaluate their medical benefit.”.
Ln 85-92 See general comments. Much more detail of procedures used is needed or if based on literature this must be cited. Were extractions really done for one month? Under what conditions?
We thank the Reviewer for his/her suggestion and, according to his/her indications, we have improved the methods section concerning plant extracts preparation, as follows:
2.2. Plant extracts. Epilobium parviflorum, Melilotus officinalis and Cardiospermum halicacabum were kindly provided by Agripharma agricultural cooperative society (Padua, Italy). In detail, Epilobium parviflorum (Schreb.) (collected plant material from North Europe; voucher No.: BPLR070ATXA), Melilotus officinalis and Cardiospermum halicacabum (cultivated plant material from Italy; voucher No.: L. MEL1809B and L. CARDI1806L, respectively) were studied. The dried aerial part of Epilobium parviflorum, aerial flower part of Melilotus officinalis and flowering tops of Cardiospermum halicacabum, containing the main active constituents of the plants, as from literature data [27-29], were obtained through low-temperature drying. Then, they were shredded and then macerated in 40% v/v ethanol, or in hot or cold glycerate with euxil 9010, for 21 days, at room temperature, at dark conditions. A ratio of 1:10 and 1:3 (g over solvent volume, ml) was used for 40% v/v ethanol and hot/cold glycerate extracts, respectively. Then, the thick mass of 40% v/v ethanol extracts was filtered several times through tangential flow microfiltration with ceramic filter, having porosity of 0.2 µm diameter, while hot or cold glycerate extracts through a paper filter with porosity of 8-20 µm diameter. Finally, the obtained liquid part, about 90%, was bottled at cold temperatures.
Ln 88-89 Why were the aerial part of Epilobium, aerial flower part of Melilotus and flowering tops of Cardiospermum used for extraction? Are these the plant parts used in folk medicine?
The referee is correct, these plant parts are widely used in folk medicine and as indicated by literature data they contain the main active constituents of the plants. We thank the Reviewer for his/her suggestion and, according to his/her indications, we have improved the manuscript by adding the following references in section 2.2. of methods:
[27]. Granica, S.; Piwowarski, J.P.; Czerwińska, M.E.; Kiss, A.K. Phytochemistry, pharmacology and traditional uses of different Epilobium species (Onagraceae): a review. J. Ethnopharmacol. 2014;156, 316-346.
[28]. Horváth, G.; Csikós, E.; Andres, E.V.; Bencsik, T.; Takátsy, A.; Gulyás-Fekete, G.; Turcsi, E.; Deli, J.; Szőke, É.; Kemény, Á.; Payrits, M.; Szente, L.; Kocsis, M.; Molnár, P.; Helyes, Z. Analyzing the Carotenoid Composition of Melilot (Melilotus officinalis (L.) Pall.) Extracts and the Effects of Isolated (All-E)-lutein-5,6-epoxide on Primary Sensory Neurons and Macrophages. Molecules, 2021; 26, 503.
[29]. Fai, D.; Fai, C.; Di Vito, M.; Martini, C.; Zilio, G.; De Togni, H. Cardiospermum halicacabum in atopic dermatitis: Clinical evidence based on phytotherapic approach. Dermatol Ther. 2020; 33, e14519.
Ln 94 ‘27Singleton and Rossi (1965)’. The number has carried over with reference.
Done
Ln 110 ‘28Broadhurst and Jones (1978)’. As above.
Done
Ln 121 1% antibiotics as in ln 119?
Yes, we have corrected accordingly
Ln 129-133 Were extracts added along with LPS or after LPS activation?
Extracts were added 30 min before LPS. We have added the sentence in the text.
Ln 133 How long before the study was media changed?
The medium was changed with serum-free immediately before cells treatments.
Ln 241 ‘ethanol, we’. Substitute ‘ethanol extract, we’.
Done
Ln 244 ‘RAW cells, any toxicity’. Substitute ‘RAW cells, no toxicity’
Done
Ln 260-262 Where is data for extracts alone. Not in Figures 1A and B, which is for LPS treated. Move ref to Figure 1A and B further down text and leave text on extracts alone as a statement (data not shown).
Data for extracts alone are reported in Figures 1A and B, showing CTR (cells alone), LPS (cells added with LPS 1ug/ml) and plant extracts alone at the concentration of 1ug/ul, E is epilobium, C cardiospermum and M melilotus.
Ln 268 ‘investigated are more potent’. investigated tend to be more potent.
Done
Ln 283-284 Where is data for extracts alone. Not in figure 2A and 2B, which is for LPS treated. Move ref to Figure 2A and 2B further down text and leave text on extracts alone as a statement (data not shown).
Data for extracts alone are reported in Figures 2A and B, showing CTR (cells alone), LPS (cells added with LPS 1ug/ml) and plant extracts alone at the concentration of 1ug/ul, E is epilobium, C cardiospermum and M melilotus.
Ln 341-344 References for this observation?
We have added references [27-29] in the Methods and in the Discussion sections.
Ln 402 ‘to contrast inflammation’ ‘to counteract inflammation’?
Counteract is fine, thank you.
Reviewer 2 Report
The presented manuscript by Stefania Merighi et al. reports antioxidant and in vitro anti-inflammatory effects of Epilobium, Melilotus and Cardiospermum plant extracts. Those species are widely used in folk medicine because of their broad range of biological activities. Thus, despite the already existing information about the anti-inflammatory properties of the species, pharmacological data supporting their therapeutic application alongside clinical research are required to evaluate its medical benefit. The authors should however clarify some aspects because the manuscript needs serious corrections.
1- Please use scientific binomial nomenclature for plants name and write it in italics throughout the text.
2- Please replace the keywords already present in the title and add more appropriate ones, such as the A2A adenosine receptor etc...
3- Section 2.1. Materials. Please shift information on Nitrate/Nitrite Colorimetric Assay Kit to the dedicated paragraph 2.11.
4- Section 2.2. Plant extract. This section is too general and approximate, the reviewer expects to find a detailed extraction method that reports information on the plant matrix (identification, voucher specimen with accession numbers, drying method), the quantity of plant matrix and volume of solvent used (ratio), the quantity of extract obtained including all the information useful for the reproducibility of the extract. Please also reports the preservatives used and include a description of the modern Microfiltration system mentioned.
5- Section 2.3. Total phenolic content, 2.4. Flavonoid content, Total condensed tannins. The authors report the results for these sections as mg (gallic acid or catechin) equivalents / g of plants, maybe you mean equivalents / g of extract? Since you used 25 μL of plant extract for each assay, it means that you used 40g of plant material for each ml of solvent, is it correct? Please clarify this point.
6- Line 94,110 and 123. Please adapt the reference format to the journal's requirements.
7- Please provide the origin cells collection of the used cell line RAW 264.7 and CHO.
8- Section 2.7. Please describe unequivocally how the various treatments of the plant extracts were done (pre-treatment or post-treatment) including LPS and especially which extract was used in the cell assay. Specify throughout the text including also the section of the discussion, which extracts showed the effects reported. Is the extract a dry extract or a tincture?
9- Table 1., Table 2 and 3 Please draw up the tables according to the format of the journal.
10- Section 3.2. Antioxidant properties of plant extracts. For a better understanding of the results obtained, please report in table 2 the IC50 values for the DPPH assay.
11- Have you considered that the preservatives added in glycerate can interfere with the results obtained from the DPPH assay?
11- Figure 1, 2 and 3 In the legends specify the extract used
12- In the discussion section, appropriate references are needed. for example, lines 344, 380 etc...
13- In the discussion section, you report fractions instead of plant extracts.
Author Response
Comments and Suggestions for Authors
The presented manuscript by Stefania Merighi et al. reports antioxidant and in vitro anti-inflammatory effects of Epilobium, Melilotus and Cardiospermum plant extracts. Those species are widely used in folk medicine because of their broad range of biological activities. Thus, despite the already existing information about the anti-inflammatory properties of the species, pharmacological data supporting their therapeutic application alongside clinical research are required to evaluate its medical benefit. The authors should however clarify some aspects because the manuscript needs serious corrections.
We thank the reviewer for his/her kind comments and suggestions to improve our work.
- Please use scientific binomial nomenclature for plants name and write it in italics throughout the text.
Done
- Please replace the keywords already present in the title and add more appropriate ones, such as the A2A adenosine receptor etc...
Done
- Section 2.1. Materials. Please shift information on Nitrate/Nitrite Colorimetric Assay Kit to the dedicated paragraph 2.11.
Done
4- Section 2.2. Plant extract. This section is too general and approximate, the reviewer expects to find a detailed extraction method that reports information on the plant matrix (identification, voucher specimen with accession numbers, drying method), the quantity of plant matrix and volume of solvent used (ratio), the quantity of extract obtained including all the information useful for the reproducibility of the extract. Please also reports the preservatives used and include a description of the modern Microfiltration system mentioned.
According to the Reviewer’s indications, we have improved the methods section concerning plant extracts preparation, as follows:
2.2. Plant extracts. Epilobium parviflorum, Melilotus officinalis and Cardiospermum halicacabum were kindly provided by Agripharma agricultural cooperative society (Padua, Italy). In detail, Epilobium parviflorum (Schreb.) (collected plant material from North Europe; voucher No.: BPLR070ATXA), Melilotus officinalis and Cardiospermum halicacabum (cultivated plant material from Italy; voucher No.: L. MEL1809B and L. CARDI1806L, respectively) were studied. The dried aerial part of Epilobium parviflorum, aerial flower part of Melilotus officinalis and flowering tops of Cardiospermum halicacabum, containing the main active constituents of the plants, as from literature data [27-29], were obtained through low-temperature drying. Then, they were shredded and then macerated in 40% v/v ethanol, or in hot or cold glycerate with euxil 9010, for 21 days, at room temperature, at dark conditions. A ratio of 1:10 and 1:3 (g over solvent volume, ml) was used for 40% v/v ethanol and hot/cold glycerate extracts, respectively. Then, the thick mass of 40% v/v ethanol extracts was filtered several times through tangential flow microfiltration with ceramic filter, having porosity of 0.2 µm diameter, while hot or cold glycerate extracts through a paper filter with porosity of 8-20 µm diameter. Finally, the obtained liquid part, about 90%, was bottled at cold temperatures.
5- Section 2.3. Total phenolic content, 2.4. Flavonoid content, Total condensed tannins. The authors report the results for these sections as mg (gallic acid or catechin) equivalents / g of plants, maybe you mean equivalents / g of extract? Since you used 25 μL of plant extract for each assay, it means that you used 40g of plant material for each ml of solvent, is it correct? Please clarify this point.
We thank the reviewer for his/her question that allows us to clarify this point. We have changed the data presented in table 1 by expressing total phenolic, flavonoid and condensed tannins content values as µg of gallic acid or catechin equivalents/µl of plant extract.
6- Line 94,110 and 123. Please adapt the reference format to the journal's requirements.
Done
7- Please provide the origin cells collection of the used cell line RAW 264.7 and CHO.
The origin of these cells was IZSLER Biobank, Brescia Italy and American Tissue Culture Collection, ATCC, Manassas, VA for RAW 264.7 and CHO, respectively. We have added this information in methods 2.6.
8- Section 2.7. Please describe unequivocally how the various treatments of the plant extracts were done (pre-treatment or post-treatment) including LPS and especially which extract was used in the cell assay. Specify throughout the text including also the section of the discussion, which extracts showed the effects reported. Is the extract a dry extract or a tincture?
We have specified that cells were treated with plant extracts for 30 min before LPS, and we have added this information in the text, section 2.7. We have also specified which extract was used in the cell assay throughout the text. The extract is a dry extract not a tincture.
9- Table 1., Table 2 and 3 Please draw up the tables according to the format of the journal.
Done.
10- Section 3.2. Antioxidant properties of plant extracts. For a better understanding of the results obtained, please report in table 2 the IC50 values for the DPPH assay.
We thank the referee for this request and we have added the IC50 values in table 2. Then, we have added the following sentence in methods section 2.8. “The IC50 values were calculated as the concentration of sample required to scavenge 50% of DPPH free radicals.”.
11- Have you considered that the preservatives added in glycerate can interfere with the results obtained from the DPPH assay?
Euxyl® PE 9010 is a preservative based on phenoxyethanol and ethylhexylglycerin. It is considered to be a broad spectrum preservative effective against both Gram-positive and Gram-negative bacteria, yeast and mold. We did not find literature data reporting any antioxidant effects by himself. Therefore, we did not test it alone in the DPPH assay. However, if the Reviewer thinks that this is necessary we are ready to do it.
11- Figure 1, 2 and 3 In the legends specify the extract used
Done
12- In the discussion section, appropriate references are needed. for example, lines 344, 380 etc…
We thank the Reviewer for his/her suggestion and, according to his/her indications, we have improved the manuscript by adding the following references:
Granica, S.; Piwowarski, J.P.; Czerwińska, M.E.; Kiss, A.K. Phytochemistry, pharmacology and traditional uses of different Epilobium species (Onagraceae): a review. J. Ethnopharmacol. 2014;156, 316-346.
Horváth, G.; Csikós, E.; Andres, E.V.; Bencsik, T.; Takátsy, A.; Gulyás-Fekete, G.; Turcsi, E.; Deli, J.; Szőke, É.; Kemény, Á.; Payrits, M.; Szente, L.; Kocsis, M.; Molnár, P.; Helyes, Z. Analyzing the Carotenoid Composition of Melilot (Melilotus officinalis (L.) Pall.) Extracts and the Effects of Isolated (All-E)-lutein-5,6-epoxide on Primary Sensory Neurons and Macrophages. Molecules, 2021; 26, 503.
Fai, D.; Fai, C.; Di Vito, M.; Martini, C.; Zilio, G.; De Togni, H. Cardiospermum halicacabum in atopic dermatitis: Clinical evidence based on phytotherapic approach. Dermatol Ther. 2020; 33, e14519.
Picón-Pagès, P.; Garcia-Buendia, J.; Muñoz, F.J. Functions and dysfunctions of nitric oxide in brain. Biochim. Biophy.s Acta Mol. Basis Dis. 2019; 1865, 1949-1967.
Calabrese, V.; Mancuso, C.; Calvani, M.; Rizzarelli, E.; Butterfield, D.A.; Giuffrida Stella, A.M. Nitric oxide in the central nervous system: Neuroprotection versus neurotoxicity. Nat. Rev. Neurosci. 2007, 8, 766–775.
13- In the discussion section, you report fractions instead of plant extracts.
OK substituted
Round 2
Reviewer 1 Report
The authors have successfully dealt with all matters raised in the initial review.
Reviewer 2 Report
All the concerns have been addressed. The present form of the manuscript is improved and suitable for a possible publication.